# Follow the Path: Reasoning over Knowledge Graph Paths to Improve LLM Factuality

## Abstract

We introduce **fs1**, a simple yet effective method that improves the factuality of reasoning traces by sourcing them from large reasoning models (e.g., `DeepSeek-R1`) and grounding them by conditioning on knowledge graph (KG) paths. We fine-tune eight instruction-tuned Large Language Models (LLMs) on 3.9K factually grounded reasoning traces and rigorously evaluate them on six complex open-domain question-answering (QA) benchmarks encompassing 23.9K questions. Our results demonstrate that our **fs1**-tuned model (32B parameters) consistently outperforms instruction-tuned counterparts with parallel sampling by 6-14 absolute points (pass@16). Our detailed analysis shows that **fs1** considerably improves model performance over more complex questions (requiring 3 or more hops on KG paths) and numerical answer types compared to the baselines. Furthermore, in single-pass inference, we notice that smaller LLMs show the most improvements. While prior works demonstrate the effectiveness of reasoning traces primarily in the STEM domains, our work shows strong evidence that anchoring reasoning to factual KG paths is a critical step in transforming LLMs for reliable knowledge-intensive tasks.

## 1 Introduction

Factual consistency of LLM-generated output is a requirement for critical real-world applications. LLM reasoning in the form of "thinking" has shown promising improvements in model performance on complex downstream tasks, such as mathematical reasoning and puzzle-like questions using additional compute resources during inference (e.g., test-time scaling; Wu et al., 2024; Muennighoff et al., 2025; Zhang et al., 2025). However, it remains an open question whether these reasoning techniques improve factuality, particularly for complex multi-hop QA (mQA). This task tests a model's ability to answer a question by synthesizing information from multiple pieces of evidence, often spread across different resources and requiring reasoning steps. We hypothesize that reasoning models should perform better than non-reasoning LLMs on the mQA task. To test this hypothesis, we source reasoning traces from state-of-the-art reasoning models and fine-tune several non-reasoning LLMs to attempt to induce reasoning capabilities. However, we have no guarantee that these reasoning traces from the large reasoning models are factually correct. In order to have a formal factual grounding in these traces, we condition the models on retrieved knowledge graph (KG) paths relevant to the questions. This is possible as KGs encode facts as directed, labeled graphs over entities and relations, which offers a verifiable foundation to inform each step of the reasoning process. We call our approach **fs1** (*factual* simple test-time scaling; Muennighoff et al., 2025).

We fine-tune eight different LLMs sizes on the original reasoning traces (**rt**; 3.4K samples) or on our KG-enhanced traces (**fs1**; 3.9K samples). We evaluate the fine-tuned models on six QA test sets spanning 23.9K questions, finding that fine-tuning on this amount of data can improve accuracy by 6-14 absolute points (pass@16) for a 32B parameter model across the benchmarks. A snapshot of our method is in Figure 1. This setup enables us to address our research question (**RQ**): *To what extent does grounding the reasoning processes of LLMs in KG paths enhance their factual accuracy for mQA?* To address this question, our contributions are as follows:

① We demonstrate that with test-time scaling (parallel sampling), our **fs1**-tuned Qwen2.5-32B model improves factual accuracy by 6-14 absolute points (at pass@16).

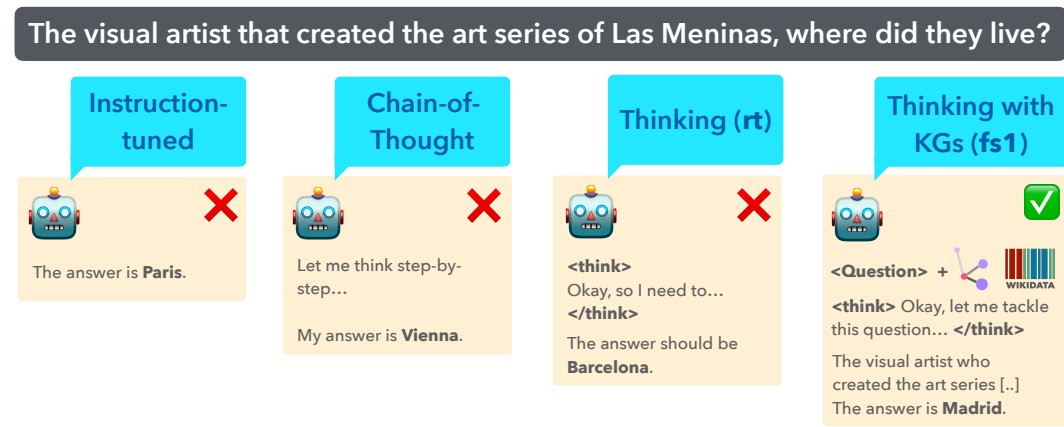

Figure 1: **Snapshot of Method.** We show a snapshot of the experiments executed in this study. There are four settings on how a question can be answered; (1) direct answer from an instruction-tuned model, (2) step-by-step reasoning via Chain-of-Thought, (3) original "thinking", and (4) knowledge-graph enhanced "thinking". We show an example of how (4) looks like in Figure 3.

②  We conduct an analysis over the question and answer types (e.g., question difficulty, answer type, and domains) to investigate where **fs1**-tuned models provide improvements. *We find that **fs1**-tuned models perform better on more difficult questions, requiring 3 hops or more.*

③  We examine performance of eight **fs1**-tuned models (360M-32B parameters) in a pass@1 setting against baselines. *We find that smaller LLMs have the largest increase in performance, whereas larger models see less profound improvements.*

④  We release 3.4K raw reasoning traces and 3.9K KG-enhanced reasoning traces both sourced from `QwQ-32B` and `Deepseek-R1`.[1]

## 2 REASONING DATA

**rt: Distilling Reasoning Traces.**    To obtain reasoning traces, we use `ComplexWebQuestions` (CWQ; Talmor & Berant, 2018), a dataset designed for complex mQA. The CWQ dataset is created by automatically generating complex SPARQL queries based on Freebase (Bollacker et al., 2008). These queries are then automatically transformed into natural language questions, which are further refined by human paraphrasing. We take the CWQ dev. set, which consists of 3,519 questions, to curate the reasoning traces. We query both `QwQ-32B` (Qwen Team, 2025) and `Deepseek-R1` (671B; DeepSeek-AI, 2025). By querying the model directly with a question, e.g., "*What art movement was Pablo Picasso part of?*", we retrieve the reasoning traces surrounded by "think" tokens (`<think>...</think>`) and force the model to give the final answer to the question in `\boxed{}` format. We extract around 3.4K correct-only traces (final answer is correct), which we call **rt**. We show full examples in Figure 8 and Figure 9 (Appendix C).

**fs1: Enhancing Reasoning Traces with Knowledge Graph Paths.**    We attempt to steer the reasoning traces with KG paths to remove the inaccuracies in the traces. Since the CWQ dataset consists of entities from Freebase, we align them to their corresponding Wikidata entities. For each question in the dev. set of the CWQ dataset, relevant KG paths are extracted from Wikidata using random walks using SPARQL queries as shown in Appendix E. Each mQA pair in the dataset may contain multiple valid KG paths, which are linearized graphs that retain the structural information of the KG. The paths are generated by extracting the relevant entities from the question and the gold answer. These diverse KG paths that can lead to the same answer reflect the possible diversity of the reasoning traces. Therefore, including linearized graphs improves the interpretability and the explainability of the reasoning traces. The prompt to obtain the improved reasoning traces is shown

---

[1]All code, datasets, and models are publicly available under an `MIT` license: https://anonymous.4open.science/r/fs1-3C18/.

Table 1: **Training Data Statistics.** Statistics of reasoning traces of `QwQ-32B` and `Deepseek-R1` on CWQ based on the `Qwen2.5-32B` tokenizer. The original reasoning traces (**rt**) are from simply querying the question to the reasoning models, whereas **fs1** indicates the statistics when queried with the knowledge graphs. We calculate the performance of the models' final answer via LLM-as-a-Judge. We show that **fs1** has higher performance in terms of accuracy compared to **rt**.

| | QwQ-32B | | R1-685B | | TOTAL | |
|---|---|---|---|---|---|---|
| | **rt** | **fs1** | **rt** | **fs1** | **rt** | **fs1** |
| Exact Match | 0.46 | ↑0.63 | 0.56 | ↑0.72 | 0.51 | ↑0.67 |
| Sem. Match (`all-MiniLM-L6-v2`) | 0.50 | ↑0.58 | 0.55 | ↑0.63 | 0.52 | ↑0.60 |
| LLM-as-a-Judge (`gpt-4o-mini`) | 0.44 | ↑0.61 | 0.54 | ↑0.70 | 0.49 | ↑0.65 |
| *Samples with only correct answers* | | | | | | |
| Number of Samples | 1,533 | 1,972 | 1,901 | 1,914 | 3,434 | 3,886 |
| Avg. Reasoning Length (subwords) | 937 | 897 | 1,043 | 637 | 990 | 767 |
| Avg. Answer Length (subwords) | 40 | 93 | 64 | 116 | 52 | 104 |

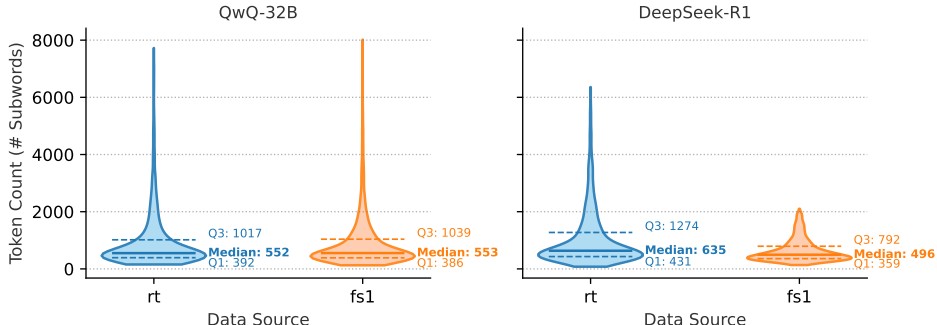

Figure 2: **Distribution of Reasoning Traces.** We show the distribution of the reasoning length among the queried models. In the left plot, we show **rt** and in the right plot, we show **fs1**. We show that particularly for **fs1** and `Deepseek-R1`, the reasoning length is shorter in terms of subwords.

in Figure 3, by prompting `QwQ-32B` and `Deepseek-R1` again. Full examples are in Figure 10 and Figure 11 (Appendix C).

**Data Statistics.** In Table 1 and Figure 2, we compare the reasoning trace accuracy and statistics of **rt** and **fs1**. We evaluate reasoning traces using three methods: (1) *Exact Match*, checking if the `\boxed{}` answer exactly matches or is a sub-phrase of any gold answer; (2) *Semantic Match*, accepting answers with a cosine similarity score >0.5; and (3) *LLM-as-a-Judge*, verifying entity alignment using `gpt-4o-mini-2024-07-18`. Results show that **fs1** achieves higher accuracy, indicating that it contains more factual answers. Traces from **rt** are longer (up to 1K subwords), **fs1** traces are typically shorter (around 800 subwords). The median length in subwords is similar for `QwQ-32B` (552 for **rt** and 553 for **fs1**), while there is a difference for `Deepseek-R1` (635 median for **rt** and 496 for **fs1**). Spot-checking reveals that **fs1** yields a more definitive answer.

## 3 METHODOLOGY

### 3.1 TRAINING AND INFERENCE

We fine-tune six `Qwen2.5-Instruct` models (0.5B to 32B) on **rt** and **fs1**, using only reasoning traces with correct final answers. During inference, we evaluate the model on the original questions to test its performance. Following Muennighoff et al. (2025), we train for 5 epochs with a sequence length of 8,192, a batch size of 16, a learning rate of $1 \times 10^{-5}$ (cosine schedule, 5% warmup), and a weight decay of $1 \times 10^{-4}$. The models are optimized with a standard supervised fine-tuning (SFT) loss, which minimizes the negative log-likelihood (implemented as the cross-entropy function) of

fs1 Prompt Example

When did the sports team owned by Leslie
    Alexander win the NBA championship?

While answering the question, make use of the
    following linearised graph as an inspiration in
    your reasoning, not as the only answer:

1994 NBA Finals, winner, Houston Rockets
Houston Rockets, owned by, Leslie Alexander
1995 NBA Finals, winner, Houston Rockets
Houston Rockets, owned by, Leslie Alexander.

Put your final answer within \boxed{}.
--
(For illustration) Gold Answer: ["1994 and 1995"]

Figure 3: **fs1 Prompt Example.** We depict how
we prompt both `Deepseek-R1` and `QwQ-32B`
to obtain better reasoning traces with KG paths.

LLM-as-a-Judge (Llama-3.3-70B)

gold answer: ["Joule per gram per kelvin", "Joule per
    kilogram per kelvin"]
predicted answer: "J/(kg $\cdot$ K)"

Is the gold answer entity or value contained in the
    predicted answer? Respond only with 0 (no)
    or 1 (yes).

# Llama–3.3–70B–Instruct outputs "1"

Figure 4: **Prompt for LLM-as-a-Judge.** We
show the LLM-as-a-Judge prompt for evaluating
whether the predicted and gold answer refer to
the same real-world entity, where *regular exact
string matching will not capture the alignment
between the gold and predicted answer* in this
example (i.e., the measurement unit).

Table 2: **Test Benchmark.** Overview of the mQA test sets used in our evaluation.

| Dataset | License | Test Size | Description |
|---|---|---|---|
| CWQ (Talmor & Berant, 2018) | `apache-2.0` | 3.5K | Multi-hop QA from WebQuestionsSP with compositional SPARQL queries for Freebase paraphrased by crowd workers. |
| ExaQT (Jia et al., 2021) | `cc-by-4.0` | 3.2K | Temporal-QA benchmark combining eight KB-QA datasets, focusing on time-specific queries. |
| GrailQA (Gu et al., 2021) | `apache-2.0` | 6.8K | Freebase QA dataset with annotated answers and logical forms (SPARQL/S-expressions) across 86 domains. |
| SimpleQA (Wei et al., 2024a) | `MIT` | 4.3K | Fact-seeking questions with verified answers, designed to measure and challenge the factual accuracy of language models. |
| Mintaka (Sen et al., 2022) | `cc-by-4.0` | 4.0K | Multilingual QA (9 languages), entity-linked pairs across diverse domains (English test split). |
| WebQSP (Yih et al., 2016) | `apache-2.0` | 2.0K | Enhanced WebQuestions with Freebase QA annotated with SPARQL ($\sim$82% coverage). |
| TOTAL | | 23.9K | |

target tokens in an autoregressive manner. Let $y_t^*$ be the correct token and $p_\theta(y_t^* \mid x, y_{<t})$ be the
model's probability of predicting it. The model optimizes the function:

$$\mathcal{L}_{\text{SFT}}(\theta) = -\frac{1}{T} \sum_{t=1}^{T} \log p_\theta \left( y_t^* \mid x, y_{<t} \right). \tag{1}$$

For inference, we use a temperature ($T$) of 0.7 and `top_p` of 0.8 for original instruct models. Other-
wise, we use $T = 0.6$ and `top_p` of 0.95. Further details on hardware and costs are in Appendix B.

## 3.2 BENCHMARKS AND EVALUATION

We show the test datasets, licenses, size and a short description in Table 2. We
have four baselines, namely `Qwen2.5-72B-Instruct` (Qwen Team, 2024), `QwQ-32B`,
`Deepseek-R1`, and `o3-mini` (OpenAI, 2025). To evaluate our models, we select a
suite of six mQA benchmarks with a total of 23.9K questions. We have four setups for
benchmarking the models: (1) All models including baselines are evaluated zero-shot (i.e.,
only querying the question); (2) the models are queried using zero-shot chain-of-thought
prompting (Kojima et al., 2022; Wei et al., 2022), where we simply append the prompt
"Put your final answer within \boxed{}. Think step-by-step."; (3) we
benchmark the models fine-tuned on **rt**; (4) we benchmark the models fine-tuned on **fs1**. In Figure 12
(Appendix D.1), we show an example of each dataset in the test benchmark.

**Possible Data Leakage.** In Figure 5, we show the overlap of the questions in the training set of
`ComplexWebQuestions` (CWQ_train) versus all the other benchmarks used in our study (all

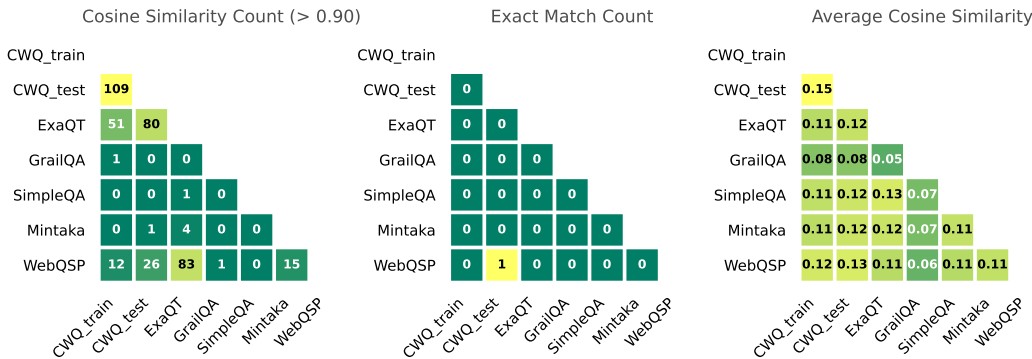

Figure 5: **Data Overlap.** We show data overlap between the train set and benchmark. On the left, one can observe the count of similar questions when the cosine similarity $>0.90$ (measured with `paraphrase-MiniLM-L6-v2`; Reimers & Gurevych, 2019). In the middle, we measure exact match counts. On the right, we show the average pairwise cosine similarity across the full test sets.

questions lower-cased). On the left, we count the times that the cosine similarity between questions exceeds 0.90. We can see that there is the most overlap between CWQ_train and CWQ_test (109 questions), and the second most is between WebQSP and ExaQT (83 questions). In the middle, we show that there is almost to none exact string match between the questions. On the right, we show the average pairwise cosine similarity across the benchmarks is lower or equal to 0.15.

**Evaluation Metric.** Similar to previous studies, e.g., Ma et al. (2025), we report pass@$k$, which reflects the probability that at least one out of $k$ randomly selected completions (drawn from a total of $n$ completions per problem) is correct. As such, it serves as an upper-bound on practical performance, which would require a subsequent selection mechanism. Formally, pass@$k$ is given by: $\mathbb{E}_{\text{problems}}\left[1 - \frac{\binom{n-c}{k}}{\binom{n}{k}}\right]$, where $n$ is the number of generated completions per problem and $c$ is the count of correct completions (Chen et al., 2021). For our benchmarks, we evaluate $k = \{1, 2, 4, 8, 16\}$. In practice, pass@32 is typically reported for formal theorem-proving tasks, while pass@1 (reducing to standard top-1 accuracy) is standard for math and coding tasks as mentioned by Ma et al. (2025). In this work for factual mQA, we report until $k = 16$.

**LLM-as-a-Judge.** To decide whether an answer is correct or not (1 or 0), our main evaluation approach is using LLM-as-a-judge with `Llama-3.3-70B-Instruct`[2] to determine whether a predicted answer obtained from the `\boxed{}` output is referring to the same real-world entity as the gold answer. An example of this is shown in Figure 4. When the model does not generate a `\boxed{}` output, we take the last 10 subwords as predicted answer, which LLM-as-a-judge can infer what the predicted real-world entity is when there is not exact string matching. This same approach is used in Table 1. Compared to exact string matching and semantic similarity evaluation methods, LLM-as-a-Judge rates the quality of output similarly compared to the other methods.

## 4 RESULTS AND DISCUSSION

### 4.1 RESULTS WITH TEST-TIME SCALING

Parallel scaling can achieve lower latency by enabling multiple (identical) models to run simultaneously locally (via e.g., multiple GPUs or batching techniques) or via API based methods to generate multiple answers. Formally, parallel sampling entails an aggregation technique that combines $N$ independent solutions into a single final prediction, commonly known as a best-of-$N$ approach (Chollet,

---

[2]We compare both `gpt-4o-mini-2024-07-18` and `Llama-3.3-70B-Instruct` on a large subsample of our outputs and saw there there is almost no difference in predictions. Additionally, `Llama-3.3-70B` is rated higher in LM Arena than `gpt-4o-mini` (at time of writing 79[th] vs. 83[rd] respectively).

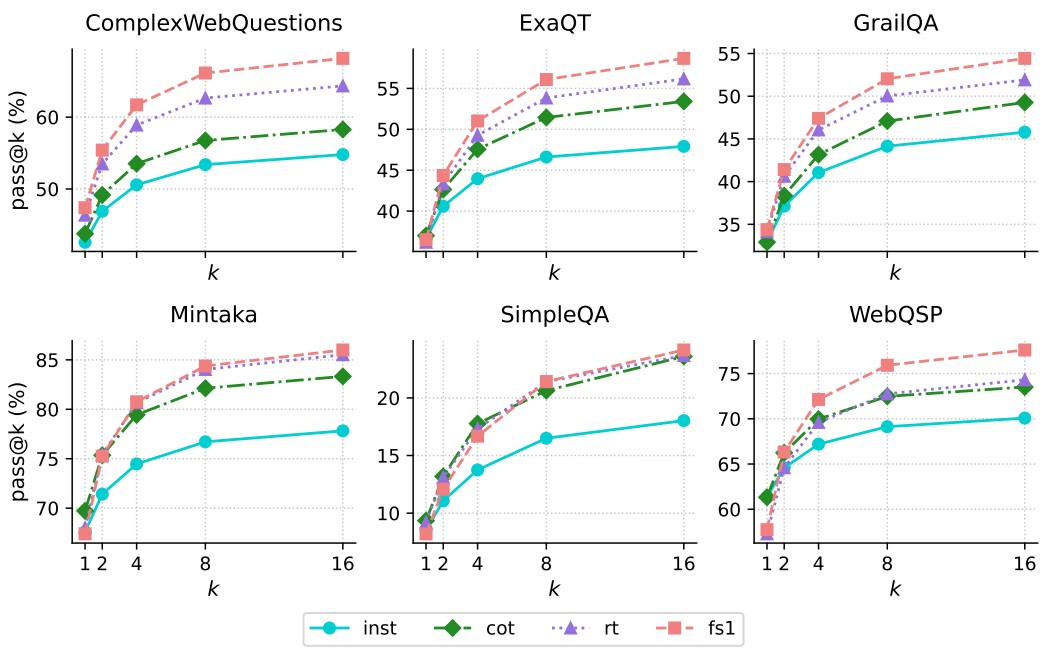

Figure 6: **Upper-bound Test-Time Scaling for Factual Reasoning.** We show with `Qwen2.5-32B` that parallel scaling is beneficial for complex mQA, measured by pass@$k$, especially when fine-tuned on **fs1**, instead of conducting single-pass inference.

2019; Irvine et al., 2023; Brown et al., 2024a; Li et al., 2022). Formally, given a set of $N$ predictions $P = \{p_1, \ldots, p_N\}$, the best-of-$N$ method selects a prediction $p \in P$ as the final output.

In this work, we present results using pass@$k$ (see Section 3.2), extending the number of sampled $k$ (until $k = 16$). In Figure 6, we show parallel scaling results by performing 16 inference runs with `Qwen2.5-32B`-Instruct, CoT, **rt**, **fs1** on each test dataset.[3] As $k$ increases, pass@$k$ (indicating whether at least one generation is correct) rises steadily across all benchmarks. Parallel sampling boosts the chance of producing a correct answer, especially when fine-tuned on **fs1**. For example, on CWQ, we see a performance increase of 16 absolute points at $k = 16$ and on SimpleQA around 6 absolute points at the same $k$ compared to their original instruction-tuned counterpart.

| CWQ_test | QwQ-32B | R1-685B |
|----------|---------|---------|
| pass@1 | 0.4549 | 0.4545 |
| pass@2 | 0.5490 | 0.5491 |
| pass@4 | 0.6223 | 0.6195 |
| pass@8 | 0.6779 | 0.6741 |
| pass@16 | 0.7195 | 0.7195 |

Table 3: **Ablation Teacher Model.** We show pass@$k$ performance of Qwen2.5-32B trained on separate subsets of **fs1** separated by teacher model applied to CWQ. We demonstrate that there is almost no difference in performance. Indicating that **fs1** is the source of improvement.

### 4.2 ARE THE GAINS COMING FROM A SUPERIOR TEACHER MODEL OR FS1?

It is not uncommon to assume that `QwQ-32B` is a weaker model than `Deepseek-R1` (685B). Therefore, it might be unclear how much the final performance gain comes from the superior reasoning of the teacher model or the factual grounding provided by the KG paths. To disentangle the effect of teacher model capabilities and **fs1**, we train Qwen2.5-32B on two separate **fs1** subsets. We take the subset of `QwQ-32B` and `Deepseek-R1` reasoning traces with the same questions (from Table 1) and fine-tune on these subsets. In Table 3, we show that there is almost no difference

---

[3]For parallel sampling, we limit ourselves to `Qwen2.5-32B` as running 16 inferences for 8 models for all 4 settings would require 12.2M model inferences for the test benchmarks, which is computationally prohibitive.

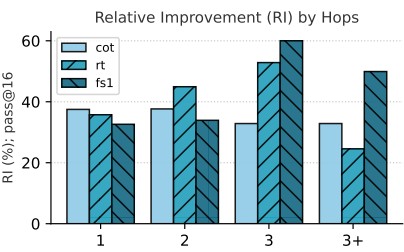
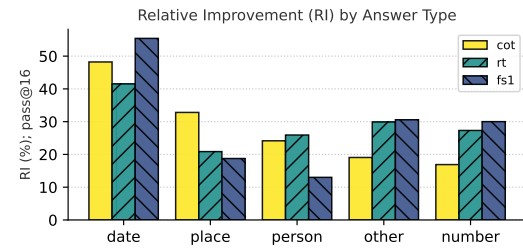

(a) Performance by number of hops required to answer the question measured in pass@16.

(b) Performance by answer type (i.e., what type of entity the answer is) in pass@16.

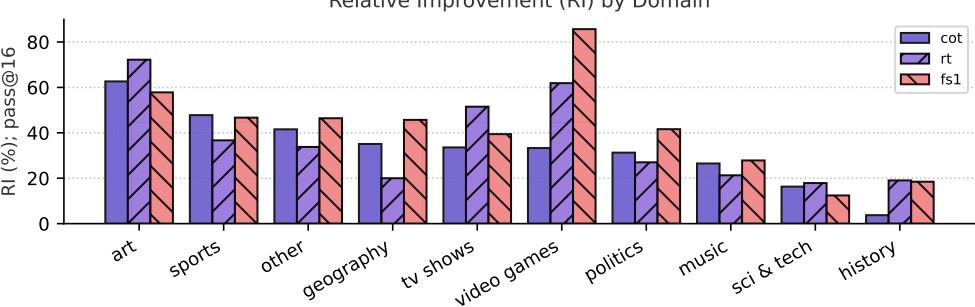

(c) Performance measured per domain in pass@16.

Figure 7: **Relative Improvements across Different Axes.** We show the relative performance improvement (%) at pass@16 of different `Qwen-32B` (i.e., CoT, **rt** and **fs1**) against the original instruct model. In (a), we show the performance of the models by the number of hops required to answer the question. In (b), we show the performance of the models by answer type. In (c), we show the performance by domain of the question. Absolute numbers are in Figure 13 (Appendix F).

in performance when using a different teacher model, when evaluated on CWQ, showing that **fs1** works as a method by using KG paths to steer the reasoning behaviour of LLMs.

### 4.3 WHAT TYPE OF SAMPLES DO MODELS SEEM TO FAIL ON?

In Figure 7, we investigate what kind of questions the model (`Qwen2.5-32B`) seems to fail on. We take metadata information from SimpleQA (Wei et al., 2024a), which indicates the question difficulty in number of hops required to answer the question (Figure 7a), type of answer (Figure 7b) and the domain of the question (Figure 7c). For question difficulty, we source the number of hops for each question in SimpleQA from Lavrinovics et al. (2025). We count the number of relations (P) from Wikidata, which would indicate the number of hops required to go from entity A to B. When a question does not contain any relations, we assume it takes more than 3 hops to answer the question.

In Figure 7a, we observe that **fs1** has lower relative improvements on easier questions (e.g., 1 or 2 hops required), but outperforms the other models when the question gets more complex (3 or more hops required). *This indicates that inducing KG paths helps answering complex questions.* In Figure 7b, we show that **fs1** has the most relative improvement on numerical answers, such as numbers, dates and also miscellaneous answer types. Last, in Figure 7c, **fs1** performs best on questions related to video games, geography, politics, music, and miscellaneous questions. Additionally, **rt** performs best on art and history-related questions. Last, CoT performs best on questions related to sports.

### 4.4 SINGLE PASS RESULTS ACROSS SCALE

In Table 4, we show results in terms of accuracy via LLM-as-a-Judge at pass@1 (i.e., one inference run per question) on all test datasets. For the baselines, we observe that o3-mini is the dominant model, achieving the highest score on five out of six datasets, such as its 0.774 accuracy on Mintaka

and 0.680 on WebQSP. The only exception is SimpleQA, where R1-70B performs best with a score of 0.188. These are followed by Qwen2.5-72B-Instruct and QwQ-32B in overall performance.

Observing the Qwen2.5 results, the benefits of fine-tuning on **rt** and **fs1** are most pronounced at the sub-billion parameter scale. For instance, fine-tuning the 0.5B model on **fs1** yields substantial relative gains across all tasks, peaking at a +74.6% on WebQSP. However, as model size increases, the performance differences become more nuanced. For the 1.5B model, the same **fs1** fine-tuning leads to performance degradation on four out of six datasets, such as ExaQT (-4.7%) and WebQSP (-1.1%). While larger models like the 32B still benefit from fine-tuning (e.g., **rt** and **fs1** are often the best performers in their group), the relative gains are smaller than those seen at the 0.5B scale.

Our results also show that fine-tuning improvements do not uniformly generalize across different model families at the sub-billion parameter scale. A comparison between the fine-tuned Qwen2.5 and SmolLM2 models reveals a significant performance divergence. Specifically, fine-tuning on **fs1** provided consistent enhancements for the Qwen2.5-0.5B model, improving its CWQ score from 0.135 to 0.209. In contrast, the same fine-tuning on SmolLM2-360M yielded mixed results; while it improved performance on most tasks, it caused a notable degradation of -15.9% on GrailQA.

This variance diminishes with scale, as models at the 1.5B/1.7B parameter scale exhibit more convergent behavior. For example, fine-tuning with **rt** on GrailQA provides a nearly identical small boost to both Qwen2.5-1.5B (+1.9%) and `SmolLM2-1.7B` (+1.8%). Overall, We hypothesize this scale-dependent effect may occur because larger models (e.g., 32B) possess stronger parametric knowledge, making them less reliant on the explicit guidance from KG paths.

## 5 RELATED WORK

Different methods that involve long chain-of-thought processes (Kojima et al., 2022; Wei et al., 2022) involving reflection, backtracking, *thinking* (e.g., DeepSeek-AI, 2025; Muennighoff et al., 2025), self-consistency (e.g., Wang et al., 2023), and additional computation at inference time, such as test-time scaling (Wu et al., 2024; Muennighoff et al., 2025; Zhang et al., 2025), have shown promising improvements in LLM performance on complex reasoning tasks. Our work intersects with efforts in factuality, knowledge graph grounding, and test-time scaling.

**Graph-enhanced In-context Learning.** Enhancing the factual consistency of LLMs using KGs has been explored in different directions, including semantic parsing methods that convert natural language questions into formal KG queries (Lan & Jiang, 2020; Ye et al., 2022). Retrieval-augmented methods (KG-RAG) (Li et al., 2023; Jiang et al., 2023; Sanmartin, 2024; Sun et al., 2024; He et al., 2024) aim to reduce LLMs' reliance on latent knowledge by incorporating explicit, structured information from a KG; reasoning on graphs (RoG) models (Luo et al., 2024) generate relation paths grounded by KGs as faithful paths for the model to follow. Sun et al. (2024) uses LLM as an agent that iteratively performs a beam search on a knowledge graph, exploring, pruning, and reasoning over multiple paths until it determines enough information has been gathered to answer the question. G-Retriever (He et al., 2024) is a RAG framework that answers questions about textual graphs by first retrieving a relevant, connected subgraph using a tree optimization formulation, and then generating a textual answer based on that subgraph. Mavromatis & Karypis (2025) might be the closest to our work, they use a graph neural net to process a dense subgraph and identify relevant answer candidate nodes, then retrieves the shortest paths connecting these candidates to the question entities, and finally provides these paths as verbalized context to an LLM for reasoning.

The earlier mentioned methods primarily focus on inference-time retrieval mechanisms, like iterative beam search (Sun et al., 2024) or subgraph optimization (He et al., 2024; Mavromatis & Karypis, 2025). Other works like Tan et al. (2025) also use KG paths to guide reasoning. Our work addresses a different aspect: We focus on improving the model's intrinsic reasoning skill. Instead of inference-time retrieval, our method uses KG paths as a one-time, offline process to create higher-quality training data. This data teaches the model to 'think' more effectively on its own.

**Long Form Factuality.** Factuality in NLP involves multiple challenges (Augenstein et al., 2024), and while prior efforts have established reasoning datasets like SAFE (Wei et al., 2024b) and SimpleQA (Wei et al., 2024a), they often lack explicit grounding in structured knowledge subgraphs.

Table 4: **Single Pass (pass@1) Results on mQA Benchmarks.** We show accuracy and relative performance gains on our benchmarks for several baselines, Qwen2.5, and SmolLM2 models. For each size, we show the original instruction-tuned model followed by versions fine-tuned with chain-of-thought, **rt**, and **fs1**. Parentheses indicate the relative improvement over the instruction-tuned counterpart. The benefits of fine-tuning are most pronounced for smaller models.

| MODEL | CWQ | ExaQT | GrailQA | SimpleQA | Mintaka | WebQSP |
|---|---|---|---|---|---|---|
| *Large Language Model Baselines* | | | | | | |
| Qwen2.5-72B | 0.481 | 0.440 | 0.361 | 0.117 | 0.736 | 0.653 |
| QwQ-32B | 0.479 | 0.390 | 0.358 | 0.097 | 0.708 | 0.612 |
| R1-70B | 0.501 | 0.476 | 0.340 | **0.188** | 0.755 | 0.549 |
| o3-mini | **0.558** | **0.497** | **0.438** | 0.138 | **0.774** | **0.680** |
| *Small Language Models (0.36B-1.7B)* | | | | | | |
| SmolLM2-360M | 0.148 | 0.088 | 0.164 | 0.024 | 0.175 | 0.235 |
| + cot | 0.151 (+2.0%) | 0.101 (+14.8%) | **0.169** (+3.0%) | 0.025 (+4.2%) | 0.188 (+7.4%) | 0.230 (-2.1%) |
| + rt | **0.192** (+29.7%) | **0.111** (+26.1%) | 0.156 (-4.9%) | **0.029** (+20.8%) | **0.202** (+15.4%) | **0.293** (+24.7%) |
| + fs1 | 0.179 (+20.9%) | 0.093 (+5.7%) | 0.138 (-15.9%) | 0.027 (+12.5%) | 0.197 (+12.6%) | 0.264 (+12.3%) |
| Qwen2.5-0.5B | 0.135 | 0.058 | 0.127 | 0.023 | 0.131 | 0.173 |
| + cot | 0.161 (+19.3%) | **0.104** (+79.3%) | 0.141 (+11.0%) | 0.031 (+34.8%) | **0.214** (+63.4%) | 0.234 (+35.3%) |
| + rt | 0.190 (+40.7%) | 0.089 (+53.4%) | 0.155 (+22.0%) | 0.022 (-4.3%) | 0.178 (+35.9%) | 0.286 (+65.3%) |
| + fs1 | **0.209** (+54.8%) | 0.101 (+74.1%) | **0.166** (+30.7%) | **0.035** (+52.2%) | 0.202 (+54.2%) | **0.302** (+74.6%) |
| Qwen2.5-1.5B | 0.234 | 0.170 | 0.208 | 0.031 | 0.316 | 0.360 |
| + cot | 0.252 (+7.7%) | **0.179** (+5.3%) | **0.216** (+3.8%) | **0.041** (+32.3%) | **0.318** (+0.6%) | **0.391** (+8.6%) |
| + rt | 0.255 (+9.0%) | 0.173 (+1.8%) | 0.212 (+1.9%) | 0.038 (+22.6%) | 0.294 (-7.0%) | 0.360 (+0.0%) |
| + fs1 | **0.263** (+12.4%) | 0.162 (-4.7%) | 0.204 (-1.9%) | 0.035 (+12.9%) | 0.301 (-4.7%) | 0.356 (-1.1%) |
| SmolLM2-1.7B | 0.248 | 0.176 | 0.219 | 0.032 | 0.293 | 0.408 |
| + cot | 0.285 (+14.9%) | 0.177 (+0.6%) | 0.209 (-4.6%) | 0.032 (+0.0%) | 0.295 (+0.7%) | 0.409 (+0.2%) |
| + rt | **0.306** (+23.4%) | **0.184** (+4.5%) | **0.223** (+1.8%) | **0.038** (+18.7%) | **0.366** (+24.9%) | **0.454** (+11.3%) |
| + fs1 | 0.305 (+23.0%) | 0.179 (+1.7%) | 0.218 (-0.5%) | 0.036 (+12.5%) | 0.341 (+16.4%) | 0.426 (+4.4%) |
| *Large Language Models (3B-32B)* | | | | | | |
| Qwen2.5-3B | 0.317 | 0.214 | 0.252 | 0.044 | 0.396 | 0.466 |
| + cot | 0.302 (-4.7%) | 0.222 (+3.7%) | 0.248 (-1.6%) | 0.048 (+9.1%) | 0.431 (+8.8%) | 0.477 (+2.4%) |
| + rt | **0.363** (+14.5%) | **0.235** (+9.8%) | **0.279** (+10.7%) | **0.053** (+20.5%) | **0.495** (+25.0%) | **0.483** (+3.6%) |
| + fs1 | 0.330 (+4.1%) | 0.205 (-4.2%) | 0.253 (+0.4%) | 0.045 (+2.3%) | 0.444 (+12.1%) | 0.406 (-12.9%) |
| Qwen2.5-7B | 0.376 | 0.281 | 0.299 | **0.070** | 0.548 | **0.580** |
| + cot | 0.383 (+1.9%) | 0.292 (+3.9%) | 0.295 (-1.3%) | 0.062 (-11.4%) | **0.580** (+5.8%) | 0.565 (-2.6%) |
| + rt | 0.401 (+6.6%) | **0.296** (+5.3%) | 0.300 (+0.3%) | 0.067 (-4.3%) | 0.576 (+5.1%) | 0.517 (-10.9%) |
| + fs1 | **0.408** (+8.5%) | 0.272 (-3.2%) | **0.303** (+1.3%) | 0.053 (-24.3%) | 0.551 (+0.5%) | 0.492 (-15.2%) |
| Qwen2.5-14B | 0.392 | 0.336 | 0.318 | 0.068 | 0.624 | **0.599** |
| + cot | 0.422 (+7.7%) | **0.356** (+6.0%) | 0.322 (+1.3%) | 0.080 (+17.6%) | 0.664 (+6.4%) | 0.592 (-1.2%) |
| + rt | 0.451 (+15.1%) | 0.352 (+4.8%) | **0.331** (+4.1%) | **0.082** (+20.6%) | **0.678** (+8.7%) | 0.562 (-6.2%) |
| + fs1 | **0.454** (+15.8%) | 0.339 (+0.9%) | 0.328 (+3.1%) | 0.079 (+16.2%) | 0.654 (+4.8%) | 0.558 (-6.8%) |
| Qwen2.5-32B | 0.428 | 0.362 | 0.334 | 0.087 | 0.674 | **0.621** |
| + cot | 0.435 (+1.6%) | **0.366** (+1.1%) | 0.332 (-0.6%) | **0.099** (+13.8%) | **0.696** (+3.3%) | 0.614 (-1.1%) |
| + rt | 0.471 (+10.0%) | **0.366** (+1.1%) | 0.342 (+2.4%) | 0.094 (+8.0%) | 0.680 (+0.9%) | 0.563 (-9.3%) |
| + fs1 | **0.477** (+11.4%) | 0.361 (-0.3%) | **0.344** (+3.0%) | 0.078 (-10.3%) | 0.682 (+1.2%) | 0.576 (-7.2%) |

In contrast, Tian et al. (2024) directly address factual accuracy by fine-tuning models on automatically generated preference rankings that prioritize factual consistency.

**Test-Time Scaling as a Performance Upper-Bound.** Our evaluation using pass@$k$ is situated within the broader context of test-time scaling, which seeks to improve performance by dedicating more compute at inference. This field encompasses parallel scaling (e.g., Best-of-N), where multiple candidate solutions are generated to increase the probability of finding a correct one (Chollet, 2019; Irvine et al., 2023; Brown et al., 2024a; Li et al., 2022), and sequential scaling, where a single solution is iteratively refined through techniques like chain-of-thought prompting and revision (Wei et al., 2022; Nye et al., 2021; Madaan et al., 2023; Lee et al., 2025; Hou et al., 2025; Huang et al., 2023; Min et al., 2024; Muennighoff et al., 2025; Wang et al., 2024b; Li et al., 2025; Jurayj et al., 2025). While practical applications of parallel scaling depend on a selection mechanism (e.g., majority voting or reward-model-based scoring) to choose the final answer (Wang et al., 2023; Christiano et al., 2017; Lightman et al., 2024; Wang et al., 2024a; Wu et al., 2024; Beeching et al., 2025; Pan et al., 2024; Hassid et al., 2024; Stroebl et al., 2024), the performance of any such method is fundamentally limited by the quality of the underlying generations, often facing diminishing returns (Brown et al., 2024b;

Snell et al., 2024; Wu et al., 2024; Levi, 2024). Our work, therefore, focuses on improving the quality of each individual reasoning trace through fine-tuning, thereby directly boosting the upper-bound potential that is measured by pass@$k$.

**Domain-specific Test-Time Scaling.**   Test-time scaling also spans specific domains like coding and medicine. Z1-7B optimizes coding tasks through constrained reasoning windows, reducing overthinking while maintaining accuracy (Yu et al., 2025). In medicine, extended reasoning boosts smaller models' clinical QA performance significantly (Huang et al., 2025), complemented by structured datasets like MedReason, which enhance factual reasoning via knowledge-graph-guided paths (Wu et al., 2025), similar to our work.

## 6 CONCLUSION

In this work, we have investigated whether grounding reasoning traces on knowledge graph paths and training models on them yield tangible gains in factual accuracy on complex open-domain QA tasks. After distilling over 3K original and knowledge-graph-enhanced reasoning traces from models such as `QwQ-32B` and `Deepseek-R1`, we fine-tuned 8 LLMs on **rt** and **fs1** and evaluated them across 6 diverse benchmarks. In short, with parallel sampling, we consistently improve 6-14 absolute points in accuracy over their instruction-tuned counterpart. Particularly, using SimpleQA, we highlight that CoT and **rt** perform better on simpler questions (1 or 2 hops required), whereas our **fs1**-tuned model performs better on more complex questions, requiring 3 hops or more. Lastly, we examined the performance of eight **fs1**-tuned models across different parameter scales, finding that smaller models (below the 1.7B parameter range) show the largest increase in performance, while larger models see less profound improvements in a pass@1 setting. By releasing all code, models, and reasoning traces, we provide a rich resource for future work on process-level verification and the development of factuality-aware reward models. In turn, we hope this work facilitates more factual large language models, making them more useful for real-world usage.

**Limitations.**   Our approach assumes that conditioning on KG paths improves the accuracy of reasoning traces, though it does not guarantee perfect intermediate processes. Additionally, accurately evaluating entity answers poses challenges; we attempted to mitigate this limitation using LLM-based judgments, but these methods have their own inherent limitations. For evaluation, we note that pass@$k$ is an upper-bound performance measure. A practical implementation would require an additional selection mechanism, such as majority voting or a verifier model, to choose the final answer. Last, some of the the test datasets used might be on the older side and English only, where we do not have control on whether the data has been included in any type of LLM pre- or post-training.

**Future Work.**   Several future research directions emerge from both the KG and test-time scaling perspectives. One promising avenue is leveraging these reasoning traces to develop process reward models, which are designed for complex reasoning and decision-making tasks where evaluating intermediate (factual reasoning) steps is critical to achieving the desired outcomes. This in fact is a crucial step towards more factual LLMs. This can be done together with KGs, one possible example could be (Amayuelas et al., 2025), where they attempt to ground every generation with a knowledge graph entry. This can possibly be done during the generation of long reasoning.

## ETHICS STATEMENT

The primary ethical motivation for this research is to enhance the factuality and reliability of LLMs. By addressing the probability of these models to generate incorrect information, our work aims to contribute positively to the development of more trustworthy AI systems. We do not foresee any direct negative ethical implications arising from this research. Instead, our goal is to provide a methodology that mitigates existing risks associated with misinformation, thereby promoting a safer and more beneficial application of language technologies.

## REPRODUCIBILITY STATEMENT

We are committed to full reproducibility. All associated artifacts, such as source code, datasets, and pretrained model weights, will be made publicly available via GitHub and the Huggingface Hub upon publication. Further details regarding the computational environment, including hardware specifications, software dependencies, and hyperparameters used for fine-tuning and inference, are documented in Section 3 and Appendix B. We acknowledge that minor variations in numerical results may arise from discrepancies in hardware or software versions; however, we have provided sufficient detail to allow for a faithful replication of our experimental setup.

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

## A   LARGE LANGUAGE MODEL USE

We made use of LLMs to polish our writing and plotting our figures.

## B   TRAINING AND INFERENCE

For running `Deepseek-R1`, `o3-mini`, and some LLM-as-a-Judge experiments with `gpt-4o-mini`, we use API-based solutions via OpenAI[4] or TogetherAI[5]. The costs of running inference on all data took around 250 USD. For fine-tuning and running inference of the local models, we make use of a large HPC cluster with hardware configurations comprising multiple nodes (depending on model size; e.g., 32B models require 4 nodes for training and 1 node for inference), each with node contains eight AMD MI250x GPU modules alongside a single 64-core AMD EPYC "Trento" CPU. The library we use for inference is `vllm` (Kwon et al., 2023). For all the experiments it resulted in around 6,500 GPU hours spent.

### B.1   ENVIRONMENTAL IMPACT

We acknowledge that conducting a large-scale analysis using LLMs comes with an environmental impact. Experiments were conducted using private infrastructure in `[Redacted]` running on green energy. A cumulative of 6,500 GPU hours of computation was performed on AMD MI250x GPU modules, which has a TDP of 500 Watts. The experiments were ran from February to September 2025. During this time, the average carbon efficiency in `[Redacted]` was 0.085 $kg/kWh$.[6] This means we released about 276 $kg$ of $CO_2$ equivalent. Estimations were conducted using the Machine Learning Impact calculator[7] presented in (Lacoste et al., 2019).

## C   EXAMPLE REASONING TRACES

We show four full examples in Figure 8 and Figure 9 for **rt** and Figure 10 and Figure 11 for fs1. For visualization purposes, we split up the specific parts in special tags: `<question>Q</question>`, `<think>T</think>`, `<gold>G</gold>` to indicate what it refers to.

## D   TEST BENCHMARK

### D.1   EXAMPLES TEST BENCHMARK

In Figure 12, we show for each dataset in the test benchmark an example of how the question and answers look like.

---

[4] https://platform.openai.com/
[5] https://api.together.ai/
[6] According to https://app.electricitymaps.com/map.
[7] Find the tool here: https://mlco2.github.io/impact.

918
919
920
921
922
923
924
925
926
927
928
929
930
931
932
933
934

---

**Deepseek-R1 Example (rt)**

<question>
In Family Guy, Seth Macfarlane plays the parent of Chris Griffin. What is his character's name?
</question>

<think>
Okay, so I need to figure out the name of the character Seth MacFarlane voices who is the parent of Chris Griffin in Family Guy. Let me start by recalling what I know about the show.\n\nFirst, Family Guy is an animated sitcom created by Seth MacFarlane. He's known for voicing several characters in the show. The main family is the Griffins. The children in the family are Chris, Meg, and Stewie, and there's also the talking dog Brian. The parents would be the father and mother of these kids.\n\nI remember that the father is Peter Griffin, who is the bumbling, overweight patriarch of the family. Seth MacFarlane does voice Peter Griffin, along with other characters like Stewie and Brian. The mother is Lois Griffin, but I think she's voiced by Alex Borstein, not Seth MacFarlane. So, since the question is about the parent of Chris Griffin that Seth MacFarlane plays, it must be Peter Griffin.\n\nWait, let me double–check. Chris is the son, so his parents are Peter and Lois. Since Seth MacFarlane voices Peter, that's the answer. But just to be thorough, are there any other parents in the show that Seth voices? For example, maybe a grandparent? But Chris's grandparents would be Peter's parents or Lois's parents. I think Peter's father is named Francis Griffin, but I don't recall if Seth voices him. Lois's parents are Carter and Babs Pewterschmidt, and they are voiced by other actors. So no, the main parent characters voiced by Seth would be Peter. \n\nAlso, Stewie is the baby, and Brian is the dog, both voiced by Seth, but they aren't parents. So yes, the answer should be Peter Griffin.
</think>

The character voiced by Seth MacFarlane who is the parent of Chris Griffin in *Family Guy* is \\boxed{Peter Griffin}.

<gold>
["Peter Griffin", "Peter Lowenbrou Griffin"]
</gold>

935
936
937
938
939
940
941
942
943
944
945
946
947
948
949
950
951
952
953
954
955
956

Figure 8: **Example Instance (rt).** A successful reasoning trace for a simple multi-hop question by `Deepseek-R1`.

957
958
959
960
961
962
963
964
965
966
967
968
969
970
971

972
973
974
975
976
977
978
979
980
981
982
983
984
985
986
987
988
989
990
991
992
993
994
995
996
997
998
999
1000
1001
1002
1003
1004
1005
1006
1007
1008
1009
1010
1011
1012

---

**QwQ-32B Example (rt)**

<question>
What Eastern Europe main religion worshipers also attend St. Mary's Cathedral, in Batticaloa, Sri Lanka?
</question>

<think>
Okay, so the user is asking about the main religion worshippers in Eastern Europe who attend St. Mary's Cathedral in Batticaloa, Sri Lanka. Hmm, first I need to break this down. Let me start by recalling that Batticaloa is a city in Sri Lanka, which is in South Asia. The cathedral mentioned is a Christian church, specifically a Catholic cathedral since it's named St. Mary's.\n\nNow, the question is about Eastern Europeans who worship there. Wait, Eastern Europe's main religions are predominantly Christian, like Eastern Orthodox or Roman Catholic in some countries. But Sri Lanka's main religions are Buddhism, Hinduism, Islam, and then there's a Christian minority. So why would Eastern Europeans be attending a cathedral there? Maybe there's a community of Eastern Europeans living or working in Batticaloa? Or perhaps it's a question about the cathedral's affiliation?\n\nAlternatively, maybe there's confusion in the question. St. Mary's Cathedral in Batticaloa is likely part of the Roman Catholic Church. Eastern Europeans who are Catholic, like from countries such as Poland, Lithuania, or parts of Croatia, might attend. But Eastern Orthodox Christians would typically go to their own churches. Since the cathedral is Catholic, the main worshippers there would be local Catholics, but the question specifies Eastern Europeans. \n\nWait, perhaps the user is thinking of the term \"Eastern Europe\" in a different context. Maybe they're referring to the Eastern Catholic Churches, which are in communion with Rome but have Eastern rites. For example, Ukrainian Catholics or Ruthenian Catholics. But would they be in Sri Lanka? Maybe there's a small expat community or missionary group. Alternatively, maybe the question is mixing up terms, and the answer is Roman Catholic, as that's the cathedral's denomination, and Eastern Europeans of that faith would attend. \n\nAlternatively, could there be a confusion with Batticaloa's demographics? The area has a significant Tamil population, but Tamils are mostly Hindu or Christian. The cathedral serves the local Catholic community, which includes Sri Lankan Tamils and others. But the question specifically asks about Eastern Europeans. Maybe the answer is Roman Catholics from Eastern Europe, but I need to confirm if there's a specific group. \n\nAlternatively, perhaps the question is a trick, and the main religion in Eastern Europe (like Orthodox) isn't the one at the cathedral, so the answer is Roman Catholics. Since the cathedral is Catholic, Eastern European Catholics would be the ones attending. So the main religion worshippers there from Eastern Europe would be Roman Catholics. \n\nI think that's the most logical answer here. The cathedral is Catholic, so Eastern Europeans who are Roman Catholic would attend. The main religions in Eastern Europe include Orthodox and Catholic, so the Catholic ones from there would be the answer.
</think>

\\boxed{Roman Catholics}.

<gold>
["Catholicism", "Roman Catholic convert", "Roman Catholic"]
</gold>

---

1013
1014
1015

Figure 9: **Example Instance (rt).** A successful reasoning trace for a simple multi-hop question by `QwQ-32B`.

1016
1017
1018
1019
1020
1021
1022
1023
1024
1025

---

**QwQ-32B Example (fs1)**

<question>
What geographic region is the country that used the Ottoman Lira used?

While answering the question, make use of the following linearised graph as an inspiration in your reasoning, not as the only answer:

Konstantin Mikhailovich Bazili, place of birth, Istanbul
Istanbul, work location, Near East.

Put your final answer within \boxed{}.
</question>

<think>
Okay, let's tackle this question step by step. The user is asking about the geographic region of the country that used the Ottoman Lira. First, I need to recall what the Ottoman Lira was. From what I remember, the lira was the currency used in the Ottoman Empire. So the country in question here is the Ottoman Empire. But since the Ottoman Empire is a historical entity and not a current country, the region would refer to where the empire was located. Now, the Ottoman Empire was a large and extensive empire that existed from the 14th century until after World War I. Its heartland was in Anatolia (modern–day Turkey) and Southeast Europe. The capital was Constantinople, which is now known as Istanbul, Turkey. The empire also controlled much of the Balkans, parts of North Africa, and the Middle East.

...
(cutoff)
...

Wait, but the question is about the geographic region of the country that used the Ottoman Lira. The country is the Ottoman Empire. The geographic region that it is in is the Near East. So combining the clues provided (Istanbul being part of the Near East), the answer would be the Near East. Alternatively, if they want the specific country now, like Turkey, but since the Ottoman Empire was not Turkey, but Turkey is the successor state. The question is a bit ambiguous. However, given the clues pointing towards Near East as the work location linked to Istanbul, I think the intended answer is the Near East region. So the final answer would be Near East.
</think>

The Ottoman Lira was the currency of the Ottoman Empire, which was centered in the geographic region known historically as the Near East. This region included the empire's heartland in Anatolia (modern–day Turkey), Istanbul (then Constantinople), parts of Southeast Europe, and the Middle East. The provided clues about Istanbul, the former capital, and its association with the Near East further reinforce this connection.

\\boxed{Near East}.

<gold>
["Near East"]
</gold>

---

Figure 10: **Example Instance (fs1).** A successful reasoning trace for a question by `QwQ-32B`.

---

**Deepseek-R1 Example (fs1)**

<question>
What part of the world map is the Government of the Republic of Seychelles located in?

While answering the question, make use of the following linearised graph as an inspiration in your reasoning, not as the only answer:

Seychellois Creole, country, Seychelles
Seychelles, continent, Africa
Seychellois Creole, indigenous to, Seychelles
Seychelles, continent, Africa.

Put your final answer within \boxed{}.
</question>

<think>
Okay, so I need to figure out where the Government of the Republic of Seychelles is located on the world map. Let me start by recalling what I know about Seychelles. I think Seychelles is a country made up of islands. But where exactly? The user provided a linearised graph that mentions Seychelles is in the continent of Africa. Hmm, that's interesting because when I think of Africa, I usually picture the mainland, but there are also island countries associated with Africa. Let me verify that. Seychelles is an archipelago in the Indian Ocean, right? So it's a group of islands northeast of Madagascar. Since Madagascar itself is off the southeastern coast of Africa, Seychelles being northeast of there would place it in the Indian Ocean, near the African continent. The graph also mentions Seychellois Creole being indigenous to Seychelles, which makes sense as the local language. So, putting this together, the Government of the Republic of Seychelles is located in the country of Seychelles, which is part of the African continent. Even though it's an island nation, geographically and politically, it's considered part of Africa. Therefore, the answer should be that the government is located in Seychelles, which is in Africa.
</think>

The Government of the Republic of Seychelles is located in the country of Seychelles. According to the provided graph, Seychelles is part of the continent of Africa. Although geographically situated in the Indian Ocean northeast of Madagascar, it is politically and regionally classified as part of Africa.

\\boxed{Africa}

<gold>
[ "Africa", "Afrique" ]
</gold>

---

Figure 11: **Example Instance (fs1).** A successful reasoning trace for a question by `Deepseek-R1`.

**Benchmark Test Examples**

```
<cwq>
Who was the president in 2012 of the country where Unidad de Valor Constante is the chosen currency?
Answers: [ "Rafael Correa", "Rafael Correa Delgado", "Rafael Vicente Correa Delgado" ]
</cwq>

<exaqt>
what nba team won the back to back championship in 1989 and 1990?
Answer: [ "Detroit Pistons" ]
</exaqt>

<grailqa>
lonnie wheeler contributed to a book edition published by what publisher?
Answer: [ "Simon & Schuster" ]
</grailqa>

<simpleqa>
Who received the IEEE Frank Rosenblatt Award in 2010?
Answer: [ "Michio Sugeno" ]
</simpleqa>

<mintaka>
How many books are in Goosebumps?
Answer: [ "235" ]
</mintaka>

<webqsp>
where did diego velazquez die?
Answer: [ "Madrid" ]
</webqsp>
```

Figure 12: **Text Examples.** For each dataset in the benchmark, we show an example.

## E    SPARQL QUERIES

The query format to retrieve the Wikidata entities for Freebase entities is given by

```
SELECT ?wikientity
WHERE
{?wikientity wdt:P646 $FREEBASE_ENTITY}
```

The general structure of the SPARQL queries for 2-hop paths between the source and target entities are given by

```
SELECT ?p1 ?p1Label ?o1 ?o1Label ?p2 ?p2Label
WHERE
{wd:$SOURCE_ENTITY ?p1  ?o1.
?o1 ?p2 wd:$TARGET_ENTITY}
SERVICE wikibase:label { bd:serviceParam wikibase:language "[AUTO_LANGUAGE],mul,en". }
```

```
SELECT ?o1 ?o1Label ?p1 ?p1Label  ?p2 ?p2Label
WHERE
{ ?p1  ?o1.
?o1 ?p2 wd:$TARGET_ENTITY}
SERVICE wikibase:label { bd:serviceParam wikibase:language "[AUTO_LANGUAGE],mul,en". }
```

## F    ABSOLUTE PERFORMANCE RESULTS FOR ANALYSIS

We show the absolute performance numbers from Figure 7 at pass@16 in Figure 13.

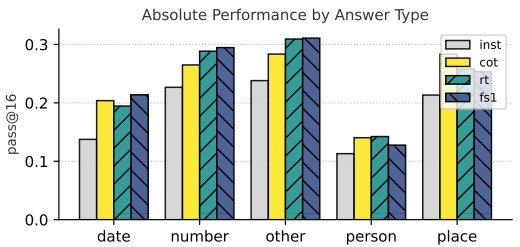

(a) Performance by number of hops required to answer the question measured in pass@16.

(b) Performance by answer type (i.e., what type of entity the answer is) in pass@16.

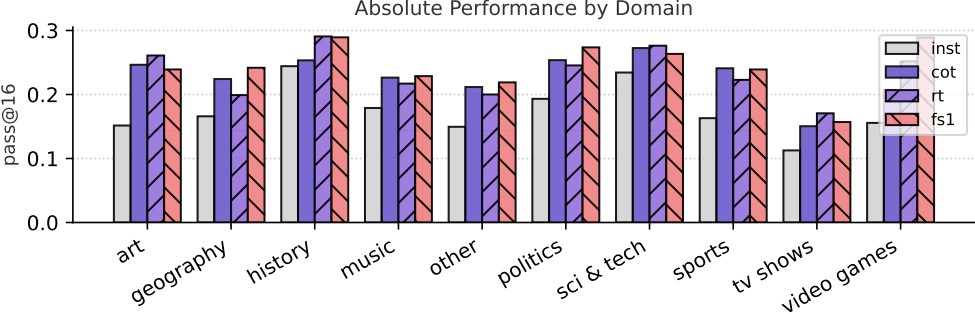

(c) Performance measured per domain in pass@16.

Figure 13: **Absolute Performance across Different Axes.** We show the absolute performance at pass@16 of different versions of `Qwen-32B` (i.e., instruct, CoT, **rt** and **fs1**). In (a), we show the performance of the models by answer type. In (b), we show the performance of the models by the number of hops required to answer the question. In (c), we show the performance by domain of the question.

