# OpenReview forum: "Follow the Path: Reasoning over Knowledge Graph Paths to Improve LLM Factuality"
_ICLR.cc/2026/Conference — ICLR 2026 Conference Withdrawn Submission_

### Official Review · Reviewer_SfuL · 2025-10-26

**Soundness:** 3
**Presentation:** 3
**Contribution:** 2
**Rating:** 4
**Confidence:** 3

**Summary:**

The paper proposes to build fine-tuning corpses by accompanying reasoning traces generated by Large Reasoning Models with reasoning paths. The paper demonstrates the effectiveness of such fine-tuning corpuses.

**Strengths:**

The proposed fine-tuning technique improves the performance of Large Language Models on several benchmarks, and some of the improvements are nontrivial.

**Weaknesses:**

The technique proposed by the paper seems essentially to be fine-tuning with both questions and contexts, with the contexts being paths mechanically extracted from knowledge graphs. Despite the performance gains, the paper will unlikely provide any new insight for the community. (Please correct me if I am wrong.)

**Questions:**

Please see “Weakness”.

---

> ### Author Response · Authors · 2025-11-20
> **Author's response**
>
> 1. The technique proposed by the paper seems essentially to be fine-tuning with both questions and contexts, with the contexts being paths mechanically extracted from knowledge graphs. Despite the performance gains, the paper will unlikely provide any new insight for the community. (Please correct me if I am wrong.)
>
> We thank the reviewer for this summary.
> While we do fine-tune with questions and KG paths, our novel contributions are:
>
> (i) A **new data curation method** that generates 'thinking paths' of demonstrably higher quality in than prior approaches (i.e., as opposed to not using it; Table 1).
>
> (ii) The key insight is that our method teaches a **generalizable reasoning skill**, not just context memorization. This is our most important finding, proven by the model's improved performance on unseen benchmarks **without** being given KG paths at test time (Fig. 6).
>
> (iii) We show this new skill is **targeted**, specifically improving complex, multi-hop reasoning (Fig. 7).
>
> Thus, our contribution is not ‘fine-tuning with context’, which we agree is not new, but a curation method that successfully imbues models with a generalizable, multi-hop reasoning ability.
>
> We thank the reviewer for their comments. With this, we hope to have addressed the reviewer’s concerns and are looking forward engaging in discussion.

---

### Official Review · Reviewer_VY98 · 2025-10-28

**Soundness:** 2
**Presentation:** 2
**Contribution:** 2
**Rating:** 4
**Confidence:** 3

**Summary:**

This paper introduces "fs1," a method to improve the factual accuracy of Large Language Models (LLMs) on complex question-answering tasks. The core idea is to fine-tune instruction-tuned LLMs on "reasoning traces" that have been grounded in knowledge graph (KG) paths. The authors first generate reasoning traces (dubbed 'rt') from powerful reasoning models like DeepSeek-R1 by prompting them with questions. To enhance the factuality of these traces, they prompt the models again, but this time conditioning the generation on relevant KG paths extracted from Wikidata.

**Strengths:**

The core concept of using KG paths to ground and improve reasoning traces is intuitive and, as the results show, effective. It's a practical approach to injecting verifiable facts into the model's reasoning process without requiring complex architectural changes.

The experimental setup is a major strength. The authors evaluate eight different model sizes across six diverse QA benchmarks, using four different prompting/fine-tuning setups (instruct, CoT, rt, fs1). This large-scale study provides robust evidence for their claims and allows for interesting insights into scaling effects. The use of LLM-as-a-Judge for evaluation is also a sensible choice for this type of open-ended generation task.

**Weaknesses:**

The fs1 traces are generated by prompting a powerful reasoning model (like DeepSeek-R1) with KG paths. It's a bit unclear how much of the final performance gain comes from the superior reasoning of the teacher model versus the factual grounding provided by the KG paths themselves. An ablation study where the reasoning traces are generated from a weaker model but still grounded with the same KG paths could help disentangle these two effects.

The process of generating the fs1 dataset involves several steps: getting the question, retrieving relevant KG paths (which itself is a non-trivial task, as hinted by the SPARQL queries in the appendix), and then prompting a very large model. How expensive and time-consuming is this data creation process? While the resulting fine-tuning dataset is small (3.9K samples), understanding the upfront cost is important for assessing the method's practicality.

**Questions:**

see weaknesses

---

> ### Author Response · Authors · 2025-11-20
> **Authors' Response**
>
> We thank reviewer VY98 for finding our method intuitive and effective. We also appreciate that the reviewer considers our experimental setup to be robust, that our results give concrete evidence for the methodology and that our evaluation is sensible.
> 1.  [..] It's a bit unclear how much of the final performance gain comes from the superior reasoning of the teacher model versus the factual grounding provided by the KG paths themselves. An ablation study where the reasoning traces are generated from a weaker model but still grounded with the same KG paths could help disentangle these two effects.
>
> This is a great suggestion, and we ran the ablation:
> | Model                     | pass@1  | pass@2  | pass@4  | pass@8  | pass@16 |
> |---------------------------|---------|---------|---------|---------|---------|
> | (fs1) Qwen2.5-32B-QwQ     | 0.4549  | 0.5490  | 0.6223  | 0.6779  | 0.7195  |
> | (fs1) Qwen2.5-32B-R1      | 0.4545  | 0.5491  | 0.6195  | 0.6741  | 0.7195  |
>
> We added a new subsection (4.2) in the updated manuscript, stating the following, given the results highlighted in blue:
> “It is not uncommon to assume that QwQ-32B is a weaker model than Deepseek-R1 (685B). Therefore, it might be unclear how much the final performance gain comes from the superior reasoning of the teacher model or the factual grounding provided by the KG paths. To disentangle the effect of teacher model capabilities and fs1, we train Qwen2.5-32B on two separate fs1 subsets. We take the subset of QwQ-32B and Deepseek-R1 reasoning traces with the same questions (from Table 1) and fine-tune on these subsets. In Table 3, we show that there is almost no difference in performance when using a different teacher model, when evaluated on CWQ, showing that fs1 works as a method by using KG paths to steer the reasoning behaviour of LLMs.”
>
> 2. The process of generating the fs1 dataset [..]. How expensive and time-consuming is this data creation process? [..], understanding the upfront cost is important for assessing the method's practicality.
>
> We thank the reviewer for this practical question. The fs1 dataset creation is a **one-time, offline process**, not a continuous inference cost, and its cost is modest. Each query (see Appendix E) for 3-hop questions takes **10.4 seconds** on the public Wikidata endpoint.
> The main computational step (generating reasoning traces) took only **20 minutes** on a single 8x GPU node. The total monetary cost to generate the dataset's 2.5M tokens (using the DeepSeek-R1 API) was **$17 USD**. We find this one-time cost to be highly practical and minimal compared to standard data curation efforts.
>
> We thank the reviewer for their comments. With this, we hope to have addressed the reviewer’s concerns and are looking forward to engaging in discussion.

---

> > ### Comment · Reviewer_VY98 · 2025-11-27
> >
> > Thanks for the reply. I will keep my score.

---

> > > ### Author Response · Authors · 2025-11-27
> > >
> > > We thank the reviewer for engaging, we believe the current score (4) is insufficiently explained solely by the these two weaknesses. We would therefore be grateful if the reviewer could expand a bit on the comment that "the score will be kept the same", so that we can better understand whether the concerns have been properly addressed and how to further strengthen the paper.

---

### Official Review · Reviewer_5bsJ · 2025-10-29

**Soundness:** 2
**Presentation:** 2
**Contribution:** 2
**Rating:** 4
**Confidence:** 4

**Summary:**

The paper investigated whether grounding reasoning traces on knowledge graph paths and training models on them yield tangible gains in factual accuracy on complex open-domain QA tasks. They first sourced the reasoning paths from large reasoning models, then grounded them into KG paths, and finally finetuned the models with these KG-enhanced reasoning paths. They provide some key findings which help us better understand when and to what extent the KG helps.

**Strengths:**

- Provide insights into when and to what extent KG-path enhanced reasoning helps LLMs to improve factual accuracy for multi-hop QA.
- The experiments are comprehensive.

**Weaknesses:**

- Numerous works have focused on KG-enhanced reasoning/retrieval on KGs to improve factual accuracy and reduce hallucination. It has been a common acknowledgment that using KGs, either as linearized graphs or as graph tokens (learned graph embeddings), can help with mQA tasks (to name a few [1-4]). Therefore, I believe this paper merely verifies a well-studied observation.
- The technical contribution is limited. Using SFT with KG-enhanced reasoning is a common practice, so the technical contribution is limited.

[1] Linhao Luo, Yuan-Fang Li, Gholamreza Haffari, and Shirui Pan. Reasoning on graphs: Faithful and interpretable large language model reasoning. ICLR 2024.

[2] Think-on-Graph: Deep and Responsible Reasoning of Large Language Model on Knowledge Graph. ICLR 2024.

[3] G-retriever: Retrieval-augmented generation for textual graph understanding and question answering. NeurIPS 2024.

[4] Gnn-rag: Graph neural retrieval for efficient large language model reasoning on knowledge graphs. ACL 2025.

**Questions:**

Will there be cases where the reasoning paths (from entities in the questions to the golden answers) are too numerous and cannot fit into the context window?

---

> ### Author Response · Authors · 2025-11-20
> **Authors' Response**
>
> We thank reviewer 5bsJ for their comments and appreciate that the reviewer found the experiments to be comprehensive and insightful to the extent that KG-paths improve factual accuracy for multihop QA, similar to reviewer VY98 and reviewer SfuL.
>
> We would like to answer the concerns point by point:
> 1. Numerous works have focused on KG-enhanced reasoning/retrieval on KGs to improve factual accuracy and reduce hallucination. [..] [1-4]
>
> We thank the reviewer for highlighting these important works [1-4], which effectively use KGs for inference-time retrieval (RAG) to improve factual accuracy.
>
> These methods, such as [1, 2, 3], primarily focus on complex, on-the-fly retrieval mechanisms, like iterative beam search [2] or subgraph optimization [3], to provide context to an LLM at inference time. [4] also uses a GNN to retrieve paths dynamically. Indeed, these RAG-based approaches are powerful for reducing hallucinations.
>
> **In contrast, our work addresses a different aspect:** we focus on improving the model's intrinsic reasoning skill. Instead of inference-time retrieval, our method uses KG paths as a **one-time, offline process** to create higher-quality training data. This data teaches the model to 'think' more effectively on its own.
>
> **We believe this is a valuable alternative, and we have now added a revised related work section, which can be seen in the updated PDF in Lines 414-428 in the blue highlighted text.**
>
> Finally, to provide some context to some extent, we present a table below that compares overlapping datasets from our work with those in [1-4] and the best-performing models.
>
> | Method / Model                | Base LLM          | CWQ          | WebQSP       |
> |------------------------------|-------------------|--------------|--------------|
> | Reasoning on Graphs (RoG) [1] | Llama2-7B         | 56.1 (F1)    | 70.8 (F1)    |
> | Think on Graph (ToG) [2]      | Llama2-70B        | 57.6 (acc)   | 68.9 (acc)   |
> | G-retriever [3]               | Llama2-7B         | —            | 73.8 (acc)   |
> | GNN-RAG [4]                   | Llama2-7B         | 60.4 (F1)    | 73.5 (F1)    |
> | fs1 (our work)                | Qwen2.5-32B       | **68.2 (acc)¹**  | **77.6 (acc)¹**  |
>
> ¹From Table 6.
>
> 2.  The technical contribution is limited. Using SFT with KG-enhanced reasoning is a common practice, so the technical contribution is limited.
>
> We agree that SFT is a standard method. Our novel contribution is not SFT itself, but our **data curation method** for KG-enhanced reasoning. We designed a novel strategy to structure data that specifically teaches the model's 'thinking' process, rather than just providing or retrieving facts at inference as references [1-4] conduct. Our results (Table 6) show this is a practical approach that yields a model with enhanced reasoning, which we believe is a valuable contribution.
>
> 3. Question: [..] Cases where reasoning paths cannot fit in context?
>
> The main model, Qwen2.5, uses a 128,000 token context window, which is substantial. For instance, for the fs1 dataset, we count the number of whitespace-separated tokens of the KG paths and find that it is on average 105 tokens long, indicating the context length of the model is more than sufficient.
>
> Overall, we thank the reviewer for their comments. With this, we hope to have addressed the reviewer’s concerns and are looking forward to engaging in discussion.

---

### Author Response · Authors · 2025-11-28
**Summary of Changes**

We thank the reviewers for their constructive feedback and positive remarks on our comprehensive experiments (5bsJ, VY98, SfuL) and robust, intuitive methodology (VY98, SfuL). We have addressed the three weaknesses and two questions raised by the three reviewers as follows:

> W1: [5bsJ] This has been done before (missing references).

We clarified in Lines 414-428 that unlike inference-time retrieval (RAG), our method uses KG paths as a one-time, offline process to improve the model's intrinsic reasoning skills.

> W2: [5bsJ + SfuL] SFT + KGs is not new.

We highlighted our specific contributions: (i) a novel curation method for high-quality 'thinking paths', (ii) show empirically that the model learns generalizable reasoning (performing well on unseen benchmarks without KGs at test time), and (iii) specific gains in complex multi-hop reasoning.

> W3: [VY98] Unclear if gains come from superior teacher model or proposed method.

We added an ablation study (Sec 4.2) confirming that gains stem from our method, not the teacher model's strength.

> Q1: [5bsJ] Cases where KG paths do not fit in context?

Clarified that paths average only 105 tokens, fitting easily within Qwen2.5’s 128k context.

> Q2: [VY98] Cost/Time

Detailed the efficiency of our one-time offline process (20 mins for 7k traces; ~$17 USD).

Finally, we believe these revisions and clarifications address all raised concerns, and we respectfully hope for a positive reconsideration.

---

### Note · Authors · 2026-01-02

I have read and agree with the venue's withdrawal policy on behalf of myself and my co-authors.